# Regret measure in continuous time limit for a stochastic Multi-armed bandit problem

## Abstract

We study a class of stochastic multi-armed bandit problems with a risk-sensitive regret measure within a continuous limit setting. This problem is interesting when optimizing the expected reward is not the foremost objective, and the problem horizon is long. Through scaling the state parameters, including the number of pulls and cumulative reward for each arm we study the bandit problem with infinite horizon, we delineate such risk using a Hamilton-Jacobi-Bellman equation with quadratic growth. Using this approach, we establish an explicit form of the optimal policy associated with the considered risk. As an application, we present examples where the results obtained in continuous time offer insights into the optimal policy for each case. Finally, numerical experiments confirm the theoretical results are presented.

***Keywords—*** Stochastic multi-armed bandit, Risk-sensitive regret, Hamilton-Jacobi-Bellman equation, Continuous time-limit

## 1 Introduction

Multi-armed bandit problem formalizes a fundamental trade-off between exploration (learning) and exploitation (control) by portraying an agent that sequentially chooses an action from a given set to receive an apriori unknown reward Thompson (1933); Katehakis & Veinott (1987); Lattimore & Szepesvari (2017); Slivkins et al. (2019). Conventionally, the main objective is to maximize the cumulative reward or minimize the total regret by applying a decision-making policy that maps the history of actions and rewards to future actions. This problem arises in real-world applications such as recommender systems.

In this paper, we intend to determining an explicit expression of the optimal policy where the problem infinite horizon. This goal is challenging because the Multi Armed Bandit problem will be treated as a continuous problem over time. Before delving into this inquiry, we revisit various strategies include the epsilon-greedy algorithm, a well-known and widely utilized method within the realm of reinforcement learning Sutton & Barto (2018). There are other kinds of randomized exploration, including Thompson sampling Thompson (1933), and upper confidence bound algorithms Lai & Robbins (1985) aims to solve optimally the Multi-Armed Bandit problem across diverse settings (Bayesian, Frequentist, structured bandit ...) with a finite and infinite horizon.

Concerning state-of-the-art research, one can distinguish two lines of literature that study the bandit problem at the infinite horizon limit with a shrinking gap between the arms:

- The first line of work Wager & Xu (2021) focuses on the diffusion limit; That is, as the horizon grows, the gaps between the arms shrink in $\frac{1}{\sqrt{n}}$. The studied diffusion asymptotics resolves a discrepancy between the $\Theta(\log(n))$ regret predicted by the fixed parameter, large-sample asymptotics on the one hand, and the $\Theta(\sqrt{n})$ regret from worst- case, finite-sample analysis on the other.

- The second line of work Fan & Glynn (2021) investigates Thompson sampling and shows that its dynamics evolve according to discrete versions of SDE's and stochastic ODE's as $n$ goes to infinity. Besides, the dynamics converge weakly to solutions of the corresponding SDE's and stochastic ODE's.

In both directions the sample size $n$ tends to infinity while the parameters remain fixed. They establish stochastic differential equations of the state that depends on the number of pulls and the cumulative reward of each arm proving convergence to diffusion limit.

The research questions that arise here are as follows: (i) Can we find a method to examine the optimal policy for the Multi-Armed Bandit problem in the continuous-time limit, and (ii) Can the behavior of the state and its evolution over time be formulated using differential equations?

Recently, there has been increasing interest in using differential equations to analyze machine learning algorithms, particularly in the context of optimization; Nevertheless, only a few connections between multi-armed bandits and differential equations have been established. The author in Adusumilli (2021) uses the process in Wager & Xu (2021) under the same diffusion regime to establish a partial differential equation (PDE) that governs the evolution of the Bayesian regret of a player as a function of the initial observed. The authors in Zhu et al. (2022) further generalize the PDE of Adusumilli (2021) to remain valid for regimes other than diffusion, i.e., potentially with different scaling factors meaning that that the state is multiplied by $\frac{1}{f(n)}$ such that the function $f(n)$ is the scaling factor, which, however, leads to a similar Hamilton-Jacobi-Bellman (HJB) equation. The PDE approach was primarily used for another classic online learning problem, namely, prediction with Expert Advice in Kobzar et al. (2020). That setting differs significantly from the bandit problem: The rewards for all "arms" (referred to as experts) are revealed to the player, making it a complete information problem. Thus, the author focus on super- and sub-solutions of these equations to provide lower and upper bounds on the value function, which is equivalent to providing classic upper and lower regret bounds. Inspired by the results obtained for Prediction with Expert Advice, Reference Kobzar & Kohn (2022) considers the particular case of a symmetric two-armed bandit. In that setting, the rewards of one arm can be used to estimate the mean of both arms due to the symmetry constraint, which makes it similar to the full feedback setting. The authors prove that the myopic player, i.e., the player selecting the arm with the best likelihood of being optimal, is minimax optimal. They also establish a PDE governing the leading term of the regret given the initial state. Then they study the regret growth rate depending on the shrinkage rate of the gaps. Studying the continuous-time limit is challenging not only because of the involved differential equations but also due to the varying settings of the bandit problem. Although the Bayesian setting is relatively straightforward to treat for general bandit problem classes, it depends on the prior over the bandit environment class. Besides, the main difficulty with Bayesian bandits is that computation of the optimal policy which is often intractable, especially when the horizon is large. Reference Zhu et al. (2022) addresses this difficulty. In this work, we examine an intermediate case that bridges between the Bayesian setting and the worst-case, in usual settings, the risk associated with pulling an arm is often overlooked. However, in many practical scenarios, such as financial portfolio optimization, risk is a primary concern for clients. More precisely, we study the risk-sensitive regret in the continuous-time limit for a general stochastic multi-armed bandit problem. Our contributions in this work include establishing a PDE that describes the evolution in continuous time limit for the setting under consideration, which is crucial for analyzing the theoretical approach of the studied measure of performance. The analytic and numerical study of the obtained PDE derives optimal policies in the continuous case, offering practical strategies for decision-making within this stochastic bandit framework. These policies are designed to optimize performance while accounting for the inherent risks, thereby providing valuable insights for applications in fields such as finance, engineering, and operations research. This paper concentrates on the continuous-time multi-armed bandit framework. In particular, we detail the stochastic multi-armed bandit problem. In Section 2 we introduce the specific scaling and establish the PDE associated with the new risk defined via induction in Section 3. In Section 4, we present the evolution in time of the multi-armed problem presenting an explicit form of the optimal policy. Finally, in Section 5, we present some examples and we conclude by providing remarks and potential future directions.

## 2  BANDIT SETTING

We consider a $K$-armed stochastic bandit, with which a player interacts for $N > 0$ rounds, referred to as the horizon. For each round $i$, the learner chooses an arm (action) $A^i$ from the given set $\mathcal{A} = \{a_k\}_{k=1}^K$ using a policy $\pi^i$. It then receives a reward $X^i \in \mathbb{R}$, it is a context of online learning problem with partial feedback.
The environment class $\mathcal{E}$ encompasses the underlying environment $\nu$, which determines the probability distributions of the arms' rewards. More precisely, given an environment $\nu = \{P_a^\nu, a \in \mathcal{A}\} \subset \mathcal{E}$, the reward $X^i$ follows the distribution $P_{A^i}^\nu$ with expectation $\mu_{A^i}(\nu)$.
We use $\Pi$ to denote the set of policies $\pi = \{\pi^i\}_{i=1}^N$. The policy $\pi^i$ at round $i$ is a function that maps the history $H^i = (A^1, X^1, \ldots, A^{i-1}, X^{i-1})$ to the learner's action.
We model our stochastic multi-armed bandit setting by a Markov decision process, where $\mathcal{S}$ is the set of states. At each round $i$, the state $S^i$ is defined as a vector consisting of two elements: The current accumulated reward of each arm and the number of pulls of each arm so far. Formally, $S^i := (s^i, q^i)$, where $s^i = (s_k^i)_{k=1}^K$, $q^i = (q_k^i)_{k=1}^K$, and

$$q_k^i := \sum_{j=1}^i \mathbf{1}\{A^j = a_k\}, \;\; s_k^i := \sum_{j=1}^i X^j \mathbf{1}\{A^j = a_k\}. \tag{1}$$

We use the notion of *expected regret*, denoted by $R_i(\pi, \nu, S)$, to evaluate the performance of a bandit agent that interacts with an environment $\nu$ using a policy $\pi$ over $n$ rounds. In words, the regret is the difference between the agent's collected reward and that of an oracle that always pulls an optimal arm. The learner starts from state $S = S^i = (s^i, q^i)$ at time $i$ and $S' = S^{i+1} = (s^{i+1}, q^{i+1})$ at time $i+1$. Let $a^\star \in \arg\max_{k \in [K]} \mu_{a_k}(\nu)$ be

an optimal arm. We define

$$R_i(\pi, \nu, S) := \mathbb{E}\left[\sum_{l=i+1}^{n} r(a_k^l, \nu)|\nu, s^i = s, q^i = q\right], \tag{2}$$

where $r(a_k^i, \nu) := \mu_{a^\star}(\nu) - \mu_{a_k^i}(\nu)$ is the instantaneous regret at time $i$. Since we condition on the bandit environment $\nu$ and the initial state $S = S^i = (s^i, q^i)$, the expectation is only taken over the randomness due to the arm distributions and to the policy.

In the next section, we provide an essential background information on the Bayesian and frequentist frameworks, along with an overview of risk measures.

## 2.1 FREQUENTIST- AND BAYESIAN SETTING

A sophisticated decision-making policy performs well over the bandit environment class $\mathcal{E}$ and not only in a particular environment $\nu$. Two conventional performance measures, frequentist and Bayesian settings, are described below. The *Frequentist setting* evaluates the performance of a policy $\pi$ under the environment class $\mathcal{E}$ by the worst-case regret, $R_i(\pi, \mathcal{E}, S)$, defined as

$$R_i(\pi, \mathcal{E}, S) := \sup_{\nu \in \mathcal{E}} R_i(\pi, \nu, S). \tag{3}$$

The *Bayesian setting* assumes known prior distribution $\rho$ over the environment class $\mathcal{E}$ so the Bayesian regret yields

$$R_i(\pi, \rho, S) := \int_{\mathcal{E}} R_i(\pi, \nu, S)\rho(\nu|S) = \mathbb{E}\left[R_i(\pi, \nu, S)\Big|s^i = s, q^i = q\right]. \tag{4}$$

The Bayesian regret incorporates the initial state and uses the updated prior $\rho(\cdot|S)$, which is obtained by the Bayes rule.

The Bayesian and worst-case regret definitions involve two elementary coherent risk measures, namely $e_0 := \mathbf{E}[X]$ and $e_\infty := \sup[X]$. Another important incoherent risk is $e_\lambda$ that interpolates the Bayesian and the Frequentist setting by taking the limit. Formally,

$$\lim_{\lambda \to 0} e_\lambda(X) = \frac{1}{\lambda}\log[\exp(\lambda X)] = e_0(X),$$

and

$$\lim_{\lambda \to \infty} e_\lambda(X) = \lim_{\lambda \to \infty}\frac{1}{\lambda}\log[\exp(\lambda X)] = e_\infty(X),$$

The primary objective is to identify an optimal policy that minimizes the value function when it adheres to a recursive form. A conventional objective is to utilize the interpolation above to develop the optimal policy of the worst-case regret employing the PDE approach. Reference Zhu et al. (2022) attains this objective for the Bayesian setting; nevertheless, that is not achievable in the risk-sensitive considered in this paper due to the inherited nonlinearity and the randomness in the prior. Therefore, we consider an alternative, as described in the following remark.

**Remark 1** *For the sake of readability, we omit the policy $\pi$.*
*Let $R_i(\nu, S, A)$ be the regret incurred by policy $\pi$ having started from state $S$ at time $i$. We denote $A_{i+1}$ by $A$. The regret conditioned on the event $A$ for a bandit environment $\nu$ is defined as*

$$R_i(\nu, S, A) = r_A(\nu) + \mathbb{E}\left[R_{i+1}(\nu, S')|\nu, S, A\right]. \tag{5}$$

*However, after taking the expected value of $\exp(\lambda R_i(\nu, S, A))$, equation 5 does not hold. Thus, for $V_{i+1} = \frac{1}{\lambda}\log\mathbb{E}\left[\exp \lambda R_i(\pi, \nu, S)|S\right]$, we derive an inequality employing Jensen's inequality and the Taylor expansion. That is,*

$$\mathbb{E}\left[\exp \lambda R_i(\nu, S)|S, A\right] \leq \mathbb{E}\left[\exp \lambda V_{i+1}(S')|S, A\right] + \lambda\mathbb{E}\left[r_A(\nu)\exp \lambda R_{i+1}(\nu, S')|S, A\right] + r_A(\nu),$$

$$\leq \mathbb{E}\left[\exp \lambda V_{i+1}(S')|S, A\right]\left(1 + \lambda\frac{\mathbb{E}\left[r_A(\nu)\exp \lambda R_{i+1}(\nu, S')|S, A\right]}{\mathbb{E}\left[\exp \lambda V_{i+1}(S')|S, A\right]} + r_A(\nu)\right). \tag{6}$$

*However, one cannot simplify equation 6 further because $r_A(\nu)$ and $\exp \lambda R_{i+1}(\nu, S')$ are dependent. To overcome this challenge, we choose to consider a risk that is defined via induction action value function. We provide the details in the next subsection.*

## 2.2 RISK DEFINED VIA INDUCTION ACTION VALUE FUNCTION

For an arbitrary policy $\pi$, we define the action value function as

$$U_i^\pi(a, s, q) := \mathcal{R}(a, s, q) + \frac{1}{\lambda} \log \left\{ \mathbb{E}[e^{\lambda \cdot V_{i+1}^\pi(s', q')} | s_{i+1} = s', q_{i+1} = q', a_k = a] \right\}, \ \forall \, (a, s, q) \in \mathcal{A} \times S, \tag{7}$$

where $\mathcal{R}(a, s, q) = \mu^\star - \mu_a$. We assume that the value function $V_i^\pi$ associated to the action value function given by equation 7 satisfies the required conditions to guarantee the existence of an optimal policy $\pi^\star$. The policy then yields the optimal value $V_i^\star = \min_\pi V_i^\pi(s, q)$ so that the following holds:

$$U_i^\star(a, s, q) = \mathcal{R}(a, s, q) + \frac{1}{\lambda} \log \mathbb{E}[e^{\lambda \cdot V_{i+1}^\star(s', q')} | s_{i+1} = s', q_{i+1} = q', a_k = a]], \tag{8}$$

$$V_i^\star(s, q) = \min_a U_i^\star(a, s, q), \quad V_{N+1}^\star(s, q) = 0.$$

## 3 LIMITING REGIME FOR SCALED TIME AND REGRET

In this section, we consider the case with a sufficiently long horizon to be represented by a rescaling in time. We study a limiting regime where the time is divided by $n$, resulting in continuous time indices $t \in [0, 1]$. The collected rewards are divided by $f(n)$ with $f > 0$ such that $\lim_\infty f(n) = \infty$, and the reward expectations by $\frac{n}{f(n)}$. We describe some other conditions on $f$ in the analysis. We also rescale other parameters to drive the continuous in-time limit as

$$t = \frac{i-1}{n}, \ \hat{q} = \frac{q}{n}, \ \hat{s} = \frac{s}{f(n)}, \tag{9}$$

The re-scaled action value function then yields

$$w^\pi \left( \frac{i-1}{n}, \frac{s}{f(n)}, \frac{q}{n} \right) := \frac{\mathcal{R}(a, s, q)}{f(n)} + \frac{1}{\lambda} \log \mathbb{E} \left[ e^{\lambda \cdot W^\pi \left( \frac{i}{n}, \frac{s}{f(n)}, \frac{q}{n} \right)} \right], \tag{10}$$

with the rescaled value function being $W^\pi \left( \frac{i}{n}, \frac{s}{f(n)}, \frac{q}{n} \right) = \frac{1}{f(n)} v_i^\pi(s, q)$.

**Lemma 1** *Assume the moments of the $k-$th arm with the probability measure $P_k^\nu$, i.e.,*

$$\mu_k(s, q) = \mathbb{E}[x | s^i = s, q^i = q], \quad \sigma_k(s, q) = \mathbb{E}[x^2 | s^i = s, q^i = q]$$
$$\mathbb{E}^P(s, q) = \mathbb{E}[x^p | s^i = s, q^i = q],$$

*exist and are finite. The scaled action value function $w^\pi$, given by equation 10, satisfies the following:*

$$w^\pi \left( \frac{i-1}{n}, \frac{s}{f(n)}, \frac{q}{n} \right) = \frac{\mathcal{R}(a, s, q)}{f(n)} + w^\pi \left( \frac{i}{n}, \frac{s}{f(n)}, \frac{q+e_k}{n} \right) + \frac{1}{f(n)} \frac{\mu_k(s, q)}{f(n)} \partial_{s_k} w^\pi \left( \frac{i}{n}, \frac{s}{f(n)}, \frac{q+e_k}{n} \right)$$

$$+ \frac{1}{2f(n)} \left( \partial_{s_k}^2 w^\pi \left( \frac{i}{n}, \frac{s}{f(n)}, \frac{q+e_k}{n} \right) + \frac{\lambda}{f(n)} \left( \partial_{s_k} w^\pi \left( \frac{i}{n}, \frac{s}{f(n)}, \frac{q+e_k}{n} \right) \right)^2 \right) \frac{\sigma_k^2(s, q)}{f^2(n)}$$

$$+ \left( \partial_{s_k}^p w^\pi \left( \frac{i}{n}, \frac{s}{f(n)}, \frac{q+e_k}{n} \right) \right) \frac{\mathbb{E}_k^P(s, q)}{f^p(n)}. \tag{11}$$

**Proof 1** *Let $\tilde{S}$ be the new state such that $\tilde{S} = S + (0_k, \frac{e_k}{n})$. That means $(s_{k+1}, q_{k+1}) = (s_k + 0_k, q + e_k)$, where $e_k$ is a $K$-dimensional vector with the $k-$th element being $1$ and all other elements being $0$. Besides, $0_k$ is the null vector. We start by using the Taylor expansion of the term*

$$V_{i+1}^\pi(s + xe_k, q + e_k) = \sum_{p=0}^\infty \frac{1}{p!} \partial_{s_k} V_{i+1}^\pi(s, q + e_k) x^p. \tag{12}$$

*The re-scaled value function $W^\pi \left( \frac{i-1}{n}, \frac{s}{f(n)}, \frac{q}{n} \right) = \frac{1}{f(n)} V_i^\pi((s, q))$ satisfies*

$$W^\pi \left( \frac{i-1}{n}, \frac{s}{f(n)}, \frac{q}{n} \right) = W^\pi \left( \frac{i}{n}, \frac{s}{f(n)}, \frac{q}{n} \right) + \frac{x}{f(n)} \partial_{s_k} W^\pi \left( \frac{i}{n}, \frac{s}{f(n)}, \frac{q}{n} \right)$$

$$+ \frac{x^2}{2f(n)} \partial_{s_k}^2 W^\pi \left( \frac{i}{n}, \frac{s}{f(n)}, \frac{q}{n} \right) + \sum_{p=3}^\infty \frac{1}{p!} \frac{x^p}{f(n)^p} \partial_{s_k}^p W^\pi \left( \frac{i}{n}, \frac{s}{f(n)}, \frac{q}{n} \right). \tag{13}$$

*Applying the $\exp(\lambda \cdot)$ function, we get*

$$\exp \lambda W^{\pi} = \exp\left(\lambda W^{\pi}\left(\frac{i}{n}, \frac{s}{f(n)}, \frac{q}{n}\right)\right)$$
$$\times \exp\left(\lambda \frac{x}{f(n)} \partial_{s_k} W\left(\frac{i}{n}, \frac{s}{f(n)}, \frac{q}{n}\right) + \lambda \frac{x^2}{2f(n)} \partial_{s_k}^2 W^{\pi}\left(\frac{i}{n}, \frac{s}{f(n)}, \frac{q}{n}\right)\right)$$
$$+ \lambda \sum_{p=3}^{\infty} \frac{1}{p!} \frac{x^p}{f(n)^p} \partial_{s_k}^p W^{\pi}\left(\frac{i}{n}, \frac{s}{f(n)}, \frac{q}{n}\right)\Big),$$

*We again apply the exponential Taylor expansion, which yields*

$$\exp \lambda W^{\pi} = \exp\left(\lambda W^{\pi}\left(\frac{i}{n}, \frac{s}{f(n)}, \frac{q}{n}\right)\right)\left(1 + \lambda \frac{x}{f(n)} \partial_{s_k} W^{\pi}\left(\frac{i}{n}, \frac{s}{f(n)}, \frac{q}{n}\right)\right.$$
$$\left. + \frac{\lambda}{2}\left(\partial_{s_k}^2 W^{\pi}\left(\frac{i}{n}, \frac{s}{f(n)}, \frac{q}{n}\right) + \lambda\left(\partial_{s_k} W^{\pi}\left(\frac{i}{n}, \frac{s}{f(n)}, \frac{q}{n}\right)\right)^2\right)\frac{x^2}{f(n)^2} + \lambda \sum_{p=3}^{\infty} \frac{1}{p!} \frac{x^p}{f(n)^p} \partial_{s_k}^p W^{\pi}\left(\frac{i}{n}, \frac{s}{f(n)}, \frac{q}{n}\right)\right),$$

*Hence, taking the conditional expectation over the state and the action, applying the $\frac{1}{\lambda}\log(\cdot)$ function, using the Taylor expansion $\log(1+u) = u + o(u)$, and inserting in equation 10, we arrive at Lemma 1.*

## 4 ESTABLISHMENT OF THE PDE

In this section, we establish a PDE of the action value function equation 10. The resulting partial differential equation (PDE) is the well-known nonlinear Hamilton-Jacobi-Bellman equation. This arises because of the temporal evolution of the bandit problem can be viewed as an optimal control problem Bellman (1957).

**Theorem 1** *Assume*

$$\lim_{n\to\infty} \frac{n}{f(n)} \mu_k(f(n)\hat{s}, n\hat{q}) = \hat{\mu}_k(\hat{\nu}, \hat{s}, \hat{q}); \quad \lim_{n\to\infty} \frac{n}{f(n)^2} \sigma_k^2(f(n)\hat{s}, n\hat{q}) = \hat{\sigma}_k^2(\hat{s}, \hat{q}), \text{ and} \quad (14)$$
$$\lim_{n\to\infty} \mathbb{E}_k^p(f(n)\hat{s}, n\hat{q}) = \mathbb{E}_k^p = 0,$$

*Then, the continuous limit of $w^{\pi}$ satisfies*

$$\partial_t w^{\hat{\pi}} + \mathcal{R}(a, \hat{s}, \hat{q}) + \partial_{\hat{q}_k} w^{\hat{\pi}} + \hat{\mu}_k(\hat{s}, \hat{q})\partial_{\hat{s}_k} w^{\hat{\pi}} + \frac{1}{2}\hat{\sigma}_k^2(\hat{s}, \hat{q})[\partial_{\hat{s}_k}^2 w^{\hat{\pi}}] + \underbrace{\frac{1}{2}\hat{\sigma}_k^2(\hat{s}, \hat{q})\lambda[(\partial_{\hat{s}_k} w^{\hat{\pi}})^2]}_{} = 0, \quad t < 1,$$

*with a finial condition at $t = 1$, $w^{\hat{\pi}}(1, \hat{s}, \hat{q}) = \mathcal{R}(a, \hat{s}, \hat{q}) + \frac{1}{\lambda}$.*

**Proof 2** *Let $w^{\pi}$ be the re-scaled action value function. We use equation 11 from Lemma 1, organize the terms, and set $\delta_t = \delta_q = \delta_s = \frac{1}{n}$. We then have*

$$\frac{w^{\pi}(t+\delta_t, \hat{s}, \hat{q}) - w^{\pi}(t, \hat{s}, \hat{q})}{\delta_t} = \frac{\mathcal{R}(a, \hat{s}, \hat{q})}{\delta_t f(n)}) + \frac{w^{\pi}(t+\delta_t, \hat{s}, \hat{q}+\delta_q e_k) - w^{\pi}(t+\delta_t, \hat{s}, \hat{q})}{\delta_q}$$
$$+ \frac{1}{\delta_t f(n)} \mu_k(\nu^n, f(n)\hat{s}, n\hat{q})\partial_{\hat{s}_k} w^{\pi}(t+\delta_t, \hat{s}, \hat{q}+e_k) \qquad (15)$$
$$+ \frac{1}{2}\frac{1}{\delta_t f^2(n)} \sigma_k^2(f(n)\hat{s}, n\hat{q})\left[\partial_{\hat{s}_k}^2 w^{\pi}(t+\delta_t, \hat{s}, \hat{q}+\delta_q e_k) + \lambda(\partial_{\hat{s}_k} w^{\pi}(t+\delta_t, \hat{s}, \hat{q}))^2\right]$$
$$+ \lambda \sum_{p=3}^{\infty} \frac{1}{p!} \frac{1}{\delta_t f(n)^p} \mathbb{E}_k^p(f(n)\hat{s}, n\hat{q})\partial_{\hat{s}_k}^p w^{\pi}(t+\delta_t, \hat{s}, \hat{q}+\delta_q e_k).$$

*Taking the limit $n \to \infty$, $\delta_t, \delta_q \to 0$, and assuming that the re-scaled moments of all arms satisfy the assumptions of Theorem equation 1, then the rescaled function*

$$\frac{\mathcal{R}(a, s, q)}{f(n)} + \frac{1}{\lambda}\log\left\{\mathbb{E}\left[\exp\left(\frac{\lambda}{f(n)}V^{\pi}(nt+1, f(n)\hat{s}, n\hat{q})\right)\right]\right\}$$

*converges to $w^{\hat{\pi}}$, where $w^{\hat{\pi}}$ verify equation 15 with a boundary condition $w^{\hat{\pi}}(1, \hat{s}, \hat{q})$.*

Given the quadratic term in equation 15, establishing the existence and uniqueness of the velocity solution might require additional hypotheses, different from those initially proposed in Zhu et al. (2022). However, applying the exponential transformation, we can derive a new form of the partial differential equation (PDE) without the quadratic term.

## 4.1 THE EXPONENTIALLY TRANSFORMED EQUATION

Let $\Phi^{\hat{\pi}}(t, \hat{S})$ be the exponential transformation of the value function $w^{\pi}(t, \hat{S})$, given as $\Phi^{\hat{\pi}}(t, \hat{S}) = exp(\lambda w^{\hat{\pi}}(t, \hat{S}))$. Consequently, $w^{\hat{\pi}}(t, (\hat{s}, \hat{q})) = \frac{1}{\lambda} \log(\Phi^{\hat{\pi}}(t, (\hat{s}, \hat{q})))$. We recall the expression of the limit of the scaled action value function $w^{\hat{\pi}} = \mathcal{R}(\hat{\nu}, a) + \frac{1}{\lambda} \log \left\{ \mathbb{E} \left[ \exp \left( \lambda v^{\hat{\pi}} \right) \right] \right\}$, such that $v^{\hat{\pi}} = \lim_{n \to \infty} W^{\pi}$. Then

$$\frac{1}{\lambda} \log \left\{ \mathbb{E} \left[ \exp \left( \lambda v^{\hat{\pi}} \right) \right] \right\} = \frac{1}{\lambda} \log(\Phi^{\hat{\pi}}(t, (\hat{s}, \hat{q}))) - r_k(\hat{\nu}). \tag{16}$$

Taking the partial differentiation with respect to the time and state of $w^{\pi}$, we have

$$\partial_t w^{\hat{\pi}} = \frac{1}{\lambda}(\partial_t \Phi^{\hat{\pi}}) \exp(-\lambda w^{\hat{\pi}}),$$

$$(D_{\hat{S}} w^{\hat{\pi}}) = \frac{1}{\lambda}(D_{\hat{S}} \Phi^{\hat{\pi}}) \exp(-\lambda w^{\hat{\pi}}), \tag{17}$$

$$(D_{\hat{S}} w^{\hat{\pi}})^2 = [\frac{1}{\lambda}(D_S \Phi^{\hat{\pi}}) \exp(-\lambda w^{\hat{\pi}})]^2$$

$$= \frac{1}{\lambda^2}(D_{\hat{S}} \Phi)^2 \exp(-2\lambda w^{\hat{\pi}}),$$

$$\Delta_{\hat{S}} w^{\hat{\pi}} = [\frac{1}{\lambda}(\Delta_{\hat{S}} \Phi^{\hat{\pi}}) - \lambda(D_{\hat{S}} w^{\hat{\pi}})^2] \exp(\lambda w^{\hat{\pi}})$$

$$= \frac{1}{\lambda}[(\Delta_{\hat{S}} \Phi^{\hat{\pi}}) \exp(-\lambda w^{\hat{\pi}}) - (D_{\hat{S}} \Phi^{\hat{\pi}})^2 \exp(-2\lambda w^{\hat{\pi}})].$$

Using equation 17, $\Phi^{\hat{\pi}}$ satisfies 5

$$\frac{1}{\lambda} \exp(-\lambda w^{\hat{\pi}}) \partial_t \Phi^{\hat{\pi}} + \mathcal{R}(a, \hat{s}, \hat{q}) + \frac{1}{\lambda}(1, \hat{\mu}_k) D_{\hat{S}} \Phi^{\pi} \exp(-\lambda w^{\hat{\pi}}) + \frac{1}{\lambda} \frac{1}{2}(\hat{\sigma}_k^2, 0) \Delta_{\hat{S}} \Phi^{\hat{\pi}} \exp(-\lambda w^{\hat{\pi}})$$

$$- \frac{1}{2\lambda}(\hat{\sigma}_k^2, 0)(D_{\hat{S}} \Phi^{\hat{\pi}})^2 \exp(-2\lambda w^{\hat{\pi}}) + \frac{1}{2\lambda}(\hat{\sigma}_k^2, 0)(D_s \Phi^{\hat{\pi}})^2 \exp(-2\lambda w^{\hat{\pi}}) = 0. \tag{18}$$

After simplifying the quadratic term and multiplying equation 18 by $\lambda \Phi^{\hat{\pi}}$, we obtain

$$\partial_t \Phi^{\hat{\pi}} + (1, \hat{\mu}_k) D_{\hat{S}} \Phi^{\hat{\pi}} + \frac{1}{2}(\hat{\sigma}_k^2, 0) \Delta_{\hat{S}} \Phi^{\hat{\pi}} + \mathcal{R}(a, \hat{s}, \hat{q}) \Phi^{\hat{\pi}} = 0, \ t < 1,$$

$$\Phi(1, (\hat{s}, \hat{q})) = e^{\lambda \mathcal{R}(a, \hat{s}, \hat{q}) + 1}. \tag{19}$$

Once we take the optimal value function, equation 15 can be written as

$$\partial_t w^{\star} + \min_k \left\{ \mathcal{R}_k(\hat{s}, \hat{q}) + \partial_{\hat{q}_k} w^{\hat{\pi}} + \hat{\mu}_k(\hat{s}, \hat{q}) \partial_{\hat{s}_k} w^{\hat{\pi}} + \frac{1}{2} \hat{\sigma}_k^2(\hat{s}, \hat{q}) \partial_{\hat{s}_k}^2 w^{\hat{\pi}} \right.$$

$$\left. + \frac{1}{2} \hat{\sigma}_k^2(\hat{s}, \hat{q}) \lambda \left( \partial_{\hat{s}_k} w^{\hat{\pi}} \right)^2 \right\} = 0, \quad t < 1,$$

$$w^{\star}(1, \hat{s}, \hat{q}) = \mathcal{R}(a, \hat{s}, \hat{q}). \tag{20}$$

We can write equation 20 as a non-linear PDE that is the HJB equation with quadratic growth. That is,

$$\partial_t w^{\star} + \min_{\hat{\pi}(t, \hat{s}, \hat{q})} \sum_{k=1}^{K} (\mathcal{R}_k(\hat{s}, \hat{q}) + \partial_{\hat{q}_k} w^{\pi} + \hat{\mu}_k(\hat{\nu}, \hat{s}, \hat{q}) \partial_{\hat{s}_k} w^{\pi}$$

$$+ \frac{1}{2} \hat{\sigma}_k^2(\hat{s}, \hat{q}) \partial_{\hat{s}_k}^2 w^{\hat{\pi}} + \frac{1}{2} \hat{\sigma}_k^2(\hat{s}, \hat{q}) \lambda \left( \partial_{\hat{s}_k} w^{\hat{\pi}} \right)^2 ) \hat{\pi}_k = 0, \quad t < 1,$$

$$w^{\star}(1, \hat{s}, \hat{q}) = \mathcal{R}(a, \hat{s}, \hat{q}), \tag{21}$$

such that $\hat{\pi}(t, \hat{s}, \hat{q})$ is the feedback control that corresponds to the policy in the bandit problem. The policy $\hat{\pi}(t, \hat{s}, \hat{q}) = \lim_{n \to \infty} \pi(nt + 1, f(n)\hat{s}, n\hat{q})$ and the rescaled optimal policy $\pi^{\star}(nt + 1, f(n)\hat{s}, n\hat{q})$ converge to $\hat{\pi}^{\star}(t, \hat{s}, \hat{q})$.

Note that since $\Phi^{\pi}$ and $w^{\pi}$ are related through a strictly monotone continuous transformation, an admissible (optimal) strategy for the exponentially transformed problem equation 18 is also admissible (optimal) for the risk-sensitive problem equation 15. More precisely, let $\Phi^{\star} = \min_{\hat{\pi}(t, \hat{s}, \hat{q}) \in \Delta^k} \Phi^{\hat{\pi}}$. Then the optimal policy for $\Phi^{\star}$ is also an optimal policy for $w^{\star}$, where we know $\Phi^{\star}$ satisfies the following 2-order nonlinear PDE:

$$\partial_t \Phi^{\star} + \min_{\hat{\pi}(t, \hat{s}, \hat{q})} \sum_{k=1}^{K} \left( \frac{1}{2}(\hat{\sigma}_k^2, 0) \Delta_{\hat{S}} \Phi^{\hat{\pi}} + (1, \hat{\mu}_k) D_{\hat{S}} \Phi^{\hat{\pi}} + \mathcal{R}_k(\hat{\nu}) \Phi^{\hat{\pi}} \right) \hat{\pi}_k = 0,$$

$$\Phi^{\star}(1, ((\hat{s}, \hat{q})) = e^{\lambda \mathcal{R}(a, \hat{s}, \hat{q}) + 1}. \tag{22}$$

At this step, it is crucial to investigate the well-posedness of the solution $\Phi^{\star}$ to the PDE equation 22. As it is an HJB equation with a penitential term, it has a viscosity solution, as stated in the following proposition.

**Proposition 1** *Assume $\{\hat{\mu}_k(s,q)\}_k$ and $\{\hat{\sigma}_k(s,q)\}_k$ are bounded and Lipshitz continuous in $(s,q)$. Then the exponential transformation action value function $\Phi^\pi$ is the unique viscosity solution to the HJB equation with potential equation 22.*

The limit of the optimal policy $\pi^\star(t,\hat{s},\hat{q})$ is given by

$$\pi_k^\star(t,\hat{s},\hat{q}) = \begin{cases} 1, & k = arg\,min_k\left\{\frac{1}{2}(\hat{\sigma}_k^2,0)\Delta_{\hat{S}}\Phi^{\hat{\pi}} + (1,\hat{\mu}_k)D_{\hat{S}}\Phi^{\hat{\pi}} + \mathcal{R}(a,\hat{s},\hat{q})\Phi^{\hat{\pi}}\right\} \\ \\ 0, & o.w \end{cases} \tag{23}$$

The optimal exponential transformation of the optimal value function satisfies

$$\partial_t\Phi + \frac{1}{2}(\hat{\sigma}_k^{\star 2},0)\Delta_{\hat{S}}\Phi + (1,\hat{\mu}_k^\star)D_{\hat{S}}\Phi + \mathcal{R}_k^\star(a,\hat{s},\hat{q})\Phi = 0,$$
$$\Phi(1,((\hat{s},\hat{q})) = e^{\lambda\mathcal{R}(a,\hat{s},\hat{q})+1}, \tag{24}$$

Using equation 16 as an expression of the exponential transformation, we obtain

$$\frac{1}{\lambda}\log\left\{\mathbb{E}\left[\exp\left(\lambda v^\star\right)\right]\right\} = \frac{1}{\lambda}\log(\Phi^\star(t,(\hat{s},\hat{q}))) - \mathcal{R}(a,\hat{s},\hat{q}). \tag{25}$$

Multiplying by $\lambda$ and applying the exponential function, we arrive at

$$\log\left\{\mathbb{E}\left[\exp\left(\lambda v^\star\right)\right]\right\} = \log(\Phi^\star(t,(\hat{s},\hat{q}))) - \lambda\mathcal{R}(a,\hat{s},\hat{q})$$
$$\Phi^\star = e^{\lambda\mathcal{R}(a,\hat{s},\hat{q})}\mathbb{E}\left[\exp\left(\lambda v^\star\right)\right]. \tag{26}$$

As previously mentioned, analytical solutions for the HJB equation are limited to specific instances involving state equations and cost functional. In the forthcoming section, we introduce a numerical methodology for solving the HJB equation.

## 5 EXAMPLES AND CONCLUSION

To clarify the application of the provided theory, in this section, we apply Proposition 1, more precisely equation 22, to characterize the value function associated with two specific bandit problems as time goes to infinity and establish an explicit optimal policy.

### 5.1 EXAMPLE: ANALYSIS OF THE CUMULANT GENERATING FUNCTION FOR "TWO-ARMED" BERNOULLI BANDIT PROBLEM WITH ONE DETERMINISTIC ARM $\mu_2 = \frac{1}{2}$

In this example, we consider the "Two-armed" bandit problem. The first arm has a deterministic reward $\mu_2 = \frac{1}{2}$, and the second arm follows a Bernoulli distribution. We fix the distribution $\text{Beta}(\alpha,\beta)$ for the prior $\nu$. The posterior distribution of $\nu$ follows as $\text{Beta}(\alpha + s^i, \beta + q^i - s^i)$, where $q^i$ is the cumulative reward and $s^i$ is the number of pulls of the unknown arm before round $i$. The limit of the optimal policy in this case is given by the following corollary, the steps of the proof are given in Appendix A.1 and numerical experiences are represented in Appendix B.

**Corollary 1** *Taking the parameters appearing in equation 15, $\delta_t = \delta_q = \delta_s = \frac{1}{n}$, based on the proof of Theorem 1, we derive a numerical schema that satisfies the following PDE, which corresponds to the two-armed Bernoulli Bandit setting.*

$$\partial_t\mathcal{R}_\lambda - \exp(\lambda_i^\star)\min\left\{\lambda^2\tilde{p}(\tilde{s},\tilde{q})(1-\tilde{p}(\tilde{s},\tilde{q}))(\partial_{\tilde{s}}\mathcal{R}_\lambda)^2 - (1-\tilde{p}(\tilde{s},\tilde{q}))\partial_{\tilde{q}}\mathcal{R}_\lambda\right.$$
$$\left. -\tilde{p}(\tilde{s},\tilde{q})\exp(-\lambda\tilde{p}(\tilde{s},\tilde{q}))\Delta_{\tilde{s}}\mathcal{R}_\lambda + \exp\left(\frac{-\lambda}{2}\right)\right\}\pi - \exp\left(\frac{-\lambda}{2}\right) = 0 \tag{27}$$

*Then The limit of the optimal policy $\pi^\star(t,\hat{s},\hat{q})$ is given by*

$$\pi_k^\star(t,\hat{s},\hat{q}) = \begin{cases} 1, & if\ k = \arg\min_k\left\{\lambda^2\tilde{p}(\tilde{s},\tilde{q})(1-\tilde{p}(\tilde{s},\tilde{q}))(\partial_{\tilde{s}}\mathcal{R}_\lambda)^2 - (1-\tilde{p}(\tilde{s},\tilde{q}))\partial_{\tilde{q}}\mathcal{R}_\lambda\right. \\ \\ \qquad\left. -\tilde{p}(\tilde{s},\tilde{q})\exp(-\lambda\tilde{p}(\tilde{s},\tilde{q}))\Delta_{\tilde{s}}\mathcal{R}_\lambda\right\}, \\ \\ 0, & o.w. \end{cases} \tag{28}$$

## 5.2 EXAMPLE: THE GAUSSIAN ONE-ARMED BANDIT

In this example we consider a Gaussian one armed bandit problem such that the first arm is assumed to be deterministic and the second arm is normally distributed with variance $\sigma^2$ over a time horizon $n$. In this setting, we set the prior $\nu \sim \mathcal{N}(\alpha(n), (\beta(n))^2)$.

**Corollary 2** *For $f(n)$ be such that there exist*

$$\lim_{n \to +\infty} \frac{\sigma^2(n)\alpha(n)}{f(n)\beta^2(n)} = \hat{\alpha}, \quad \lim_{n \to +\infty} \frac{\sigma^2(n)}{\beta^2(n)n} = \hat{\beta}, \quad \lim_{n \to +\infty}, \quad \lim_{+\infty} \frac{\sqrt{n}}{f(n)}\sigma(n) = \hat{\sigma}.$$

*Then $n \to +\infty$, the limit of the optimal policy $\pi^\star(t, \hat{s}, \hat{q})$ satisfies the proposition 1 such that $\hat{\mu}(s, q, \hat{\alpha}, \hat{\beta}) = \frac{\hat{\alpha}+s}{\hat{\beta}+q}$ and $\hat{\sigma}(s, q, \hat{\alpha}, \hat{\beta}) = \hat{\sigma}$.*

The proof of this corollary is following the proof of the lemma 4 in Zhu et al. (2022) considering the case of Gaussian one armed bandit problem it is not a particular case because we are considering a new rick measure

## 5.3 CONCLUSION

A Partial Differential Equation (PDE) approach can be beneficial for exploring various settings of the multi-armed bandit problem by providing a robust mathematical framework to analyze the dynamics and optimal strategies involved. By formulating the bandit problem as a PDE, researchers can utilize established analytical and numerical techniques to gain insights and make informed decisions.In this study, we focus on a Bayesian class of multi-armed bandit problems with an infinite horizon. We adopt the diffusion regime to scale the state parameters, including the number of pulls and the cumulative reward for each arm. Consequently, the problem transitions to continuous time. Our goal is to derive an explicit expression for the optimal policy as nn approaches infinity, while also considering a new risk measure that satisfies the Bellman equation.The development of a regret risk measure that adheres to the Bellman relation 7 enables us to tackle the challenge of randomness linked to the prior, especially when examining the entropic risk measure, as discussed in Remark 1. Utilizing the Bellman equation 7, we established a relationship between the regret risk at time $i$ and time $(i+1)$ 11. From this derived relation, we developed a discrete numerical scheme that provides insights and serves as an approximation of the continuous-time PDE. This result is established by Theorem 1. Next, we employed an exponential transformation to address the derived HJB equation, which includes a quadratic term. This technique allowed us to establish a simplified version of the HJB equation, ensuring the existence and uniqueness of a viscosity solution under the assumption that the mean $\mu(s, q)$ and $\sigma(s, q)$of each arm are bounded and Lipschitz continuous in $(s, q)$. Ultimately, we accomplished our objective by deriving an explicit expression for the optimal policy related to the examined class of stochastic bandit problems and their corresponding risk. The limit in time, as represented in Proposition 1, serves as the explicit approximation of the optimal policy after scaling the problem to a more general regime, rather than $\frac{1}{\sqrt{n}}$.In the final section of the paper, we demonstrate the application of our approach by presenting a classical example of a bandit problem that incorporates the new risk measure.

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

# A APPENDIX

## A.1 PROOF OF COROLLARY 1

We follow the steps below.

1. Define the cumulative reward starting from round $i$, having observed state $(s, q)$, $\omega^i(s, q)$ with $s^i = s, q^i = q$,. The regret starting from round $i$ follows as

$$r^i(s, q) = (n - i + 1) \max(\mu_1, \frac{1}{2}) - \omega^i(s, q).$$

2. Apply the exponential function with respect to a parameter $\lambda$ and establish the new PDE satisfied by the modified value function $\exp(\lambda r^i(\tilde{s}, \tilde{q}))$.

3. Consider that $\mathcal{R}^i_\lambda(\tilde{s}, \tilde{q})$ obtained after the rescaling $(\tilde{s} = \frac{1}{n}s, \tilde{q} = \frac{1}{n}q)$, as a value function and determinate its dynamic (PDE).

Following the steps cited above, We start by given the dynamic of the cumulative regret, let $\omega^i(s, q)$ be the cumulative reward starting from round $i$, i.e.,

$$\begin{cases} \omega^i_1(s, q) = p(s, q) + p(s, q)\omega^{i+1}(s+1, q+1) + (1 - p(s, q))\omega^{i+1}(s, q+1), & \text{if } A^i = a_1, \\ \omega^i_2(s, q) = \frac{1}{2} + \omega^{i+1}(s, q), & \text{if } A^i = a_2, \end{cases} \quad (29)$$

where $\mu_1$ is the expected reward of the first arm and $p(s, q) = \frac{\alpha+s}{\alpha+\beta+q}$ represents the expected reward when the first arm is pulled in round $i$. The regret starting from round $i$, $r^i(s, q)$, yields

$$\begin{cases} r^i_1(s, q) = \left[(n - i + 1) \max(\mu_1, \frac{1}{2})\right] - \omega^i_1(s, q), & \text{if } A^i = a_1, \\ r^i_2(s, q) = \left[(n - i + 1) \max(\mu_1, \frac{1}{2})\right] - \omega^i_2(s, q) & \text{if } A^i = a_2. \end{cases} \quad (30)$$

Using the expressions equation 29 and equation 30, we obtain

$$r^i_1(s, q) = (n - i + 1) \max(\mu_1, \frac{1}{2}) - \omega^i_1(s, q)$$
$$= (n - i + 1) \max(\mu_1, \frac{1}{2}) - \left(\mu_1 - \mu_1\omega^{i+1}(s+1, q+1) - (1 - \mu_1)\omega^{i+1}(s, q+1)\right), \quad (31)$$

and

$$r^i_2(s, q) = \left[(n - i + 1) \max(\mu_1, \frac{1}{2})\right] - \omega^i_2(s, q)$$
$$= (n - i + 1) \max(\mu_1, \frac{1}{2}) - \left(\frac{1}{2} + \omega^{i+1}(s, q)\right). \quad (32)$$

By applying the exponential function with the parameter $\lambda$, the exponential regret by selecting the first- or the second arm at round $i$ respectively yield

$$R^i_{\lambda,1}(s, q) = \exp(\lambda r^i_1(s, q)) = exp\left(\lambda\left[(n - i + 1) \max(\mu_1, \frac{1}{2})\right] - \lambda\omega^i_1(s, q)\right),$$

and

$$R^i_{\lambda,2}(s, q) = \exp(\lambda r^i_2(s, q)) = exp\left(\lambda\left[(n - i + 1) \max(\mu_1, \frac{1}{2})\right] - \lambda\omega^i_2(s, q)\right).$$

To simplify the notation, we define

$$\exp(\lambda(n - i + 1)\max(\mu_1, \frac{1}{2})) = \exp(\lambda^*).$$

Then,

$$\begin{cases} R^i_{\lambda,1}(s,q) = \exp(\lambda^*)\exp(-\lambda\omega^i_1(s,q)), \\ R^i_{\lambda,2}(s,q) = \exp(\lambda^*)\exp(-\lambda\omega^i_2(s,q)). \end{cases} \tag{33}$$

To derive a continuous limit in this setting, we apply the same re-scaling used in Section equation 3, i.e.,

$$t = \frac{i-1}{n}, \quad \hat{q} = \frac{q}{n}, \quad \hat{s} = \frac{s}{f(n)}. \tag{34}$$

Therefore, the re-scaled regret function and optimal exponential regret function respectively yield

$$\mathcal{R}^i_\lambda(\tilde{s}, \tilde{q}) = \frac{1}{n}R^i_\lambda(s,q) = \min\left(\frac{1}{n}R^i_{\lambda,1}(s,q), \frac{1}{n}R^i_{\lambda,2}(s,q)\right).$$

Using equation 33, $\mathcal{R}^i_\lambda(\tilde{s}, \tilde{q})$ satisfies

$$\mathcal{R}^i_\lambda(\tilde{s}, \tilde{q}) = \min\left(\mathcal{R}^i_{\lambda,1}(\tilde{s}, \tilde{q}), \mathcal{R}^i_{\lambda,2}(\tilde{s}, \tilde{q})\right) = \frac{\exp(\lambda^\star)}{n}\min\left(\exp(-\lambda\omega^i_1(s,q)), \exp(-\lambda\omega^i_2(s,q))\right), \tag{35}$$

such that

$$\mathcal{R}^i_{\lambda,1}(\tilde{s}, \tilde{q}) = \exp(\lambda^\star)\exp\left(-\lambda\left[\tilde{p}(\tilde{s}, \tilde{q}) + \tilde{p}(\tilde{s}, \tilde{q})v^{i+1}(\tilde{s} + \frac{1}{n}, \tilde{q} + \frac{1}{n}) + (1 - \tilde{p}(\tilde{s}, \tilde{q}))v^{i+1}(\tilde{s}, \tilde{q} + \frac{1}{n})\right]\right), \tag{36}$$

and

$$\mathcal{R}^i_{\lambda,2}(\tilde{s}, \tilde{q}) = \exp(\lambda^\star)exp\left(-\lambda\left(\frac{1}{n}\frac{1}{2} + v^{i+1}(\tilde{s}, \tilde{q})\right)\right). \tag{37}$$

We know

$$\mathcal{R}^{i+1}_\lambda\left(\tilde{s} + \frac{1}{n}, \tilde{q} + \frac{1}{n}\right) = \exp(\lambda^\star_i) - \lambda\exp(\lambda^\star_i)v^{i+1}\left(\tilde{s} + \frac{1}{n}, \tilde{q} + \frac{1}{n}\right), \tag{38}$$

and

$$\mathcal{R}^{i+1}_\lambda(\tilde{s}, \tilde{q} + \frac{1}{n}) = \exp(\lambda^\star_i) - \lambda\exp(\lambda^\star_i)v^{i+1}\left(\tilde{s}, \tilde{q} + \frac{1}{n}\right), \tag{39}$$

such that $\lambda^\star_i = \left(1 - \frac{(i+1)}{n} + \frac{1}{n}\right)\max(\mu_1, \frac{1}{2})$. Then, from the above expressions equation 37 and equation 36, we have

$$\mathcal{R}^i_\lambda(\tilde{s}, \tilde{q}) = \exp(\lambda^\star)\times$$

$$\min\left\{\exp\left(-\lambda\left[\tilde{p}(\tilde{s}, \tilde{q}) + \tilde{p}(\tilde{s}, \tilde{q})v^{i+1}\left(\tilde{s} + \frac{1}{n}, \tilde{q} + \frac{1}{n}\right) + (1 - \tilde{p}(\tilde{s}, \tilde{q}))v^{i+1}\left(\tilde{s}, \tilde{q} + \frac{1}{n}\right)\right]\right),\right.$$

$$\exp\left(-\lambda\left(\frac{1}{2n} + v^{i+1}(\tilde{s}, \tilde{q})\right)\right)\right\}$$

$$= \exp(\lambda^\star)\times\min\left\{\exp\left(-\lambda\tilde{p}(\tilde{s}, \tilde{q})\right)\exp\left(-\lambda\tilde{p}(\tilde{s}, \tilde{q})v^{i+1}\left(\tilde{s} + \frac{1}{n}, \tilde{q} + \frac{1}{n}\right)\right)\exp\left(-\lambda(1 - \tilde{p}(\tilde{s}, \tilde{q}))v^{i+1}\left(\tilde{s}, \tilde{q} + \frac{1}{n}\right)\right),\right.$$

$$\exp\left(-\frac{\lambda}{2n}\right)\exp\left(-\lambda v^{i+1}(\tilde{s}, \tilde{q})\right)\right\}.$$

Using the first order Taylor expansion for the terms

$$\exp(-\lambda\tilde{p}(\tilde{s}, \tilde{q}))v^{i+1}(\tilde{s} + \frac{1}{n}, \tilde{q} + \frac{1}{n}),$$

$$\exp\left(-\lambda(1 - \tilde{p}(\tilde{s}, \tilde{q}))v^{i+1}\left(\tilde{s}, \tilde{q} + \frac{1}{n}\right)\right)$$

and $\exp(-\lambda v^{i+1}(\tilde{s}, \tilde{q}))$, we obtain

$$\mathcal{R}^i_\lambda(\tilde{s}, \tilde{q}) = \exp(\lambda^\star)\times$$

$$\min\left\{\exp\left(-\lambda\tilde{p}(\tilde{s}, \tilde{q})\right)\left(1 - \lambda\tilde{p}(\tilde{s}, \tilde{q})v^{i+1}\left(\tilde{s} + \frac{1}{n}, \tilde{q} + \frac{1}{n}\right)\right)\right.$$

$$\times\left(1 - \lambda(1 - \tilde{p}(\tilde{s}, \tilde{q}))v^{i+1}\left(\tilde{s}, \tilde{q} + \frac{1}{n}\right)\right),$$

$$\exp\left(\frac{-\lambda}{2n}\right)\left(1 - \lambda v^{i+1}(\tilde{s}, \tilde{q})\right)\right\}.$$

Thus,
$$\mathcal{R}_\lambda^i(\tilde{s}, \tilde{q}) = \exp(\lambda^\star) \times$$

$$\min\{\exp(-\lambda \tilde{p}((\tilde{s}, \tilde{q}))) - \lambda \tilde{p}(\tilde{s}, \tilde{q}) \exp(-\lambda \tilde{p}(\tilde{s}, \tilde{q})) v^{i+1}\left(\tilde{s} + \frac{1}{n}, \tilde{q} + \frac{1}{n}\right) - \lambda(1 - \tilde{p}(\tilde{s}, \tilde{q})) v^{i+1}\left(\tilde{s}, \tilde{q} + \frac{1}{n}\right)$$
$$\tag{40}$$

$$+ \lambda^2 \tilde{p}(\tilde{s}, \tilde{q})(1 - \tilde{p}(\tilde{s}, \tilde{q})) v^{i+1}\left(\tilde{s} + \frac{1}{n}, \tilde{q} + \frac{1}{n}\right) v^{i+1}\left(\tilde{s}, \tilde{q} + \frac{1}{n}\right); \exp\left(\frac{-\lambda}{2n}\right) - \exp\left(\frac{-\lambda}{2n}\right) v^{i+1}(\tilde{s}, \tilde{q}).$$

Using equation 38 and equation 39, we obtain

$$\mathcal{R}_\lambda^i(\tilde{s}, \tilde{q}) = \min\left\{\exp(-\lambda \tilde{p}(\tilde{s}, \tilde{q})) + \tilde{p}(\tilde{s}, \tilde{q}) \exp(-\lambda \tilde{p}(\tilde{s}, \tilde{q}))\left[1 - \mathcal{R}_\lambda^{i+1}\left(\tilde{s} + \frac{1}{n}, \tilde{q} + \frac{1}{n}\right) \exp(-\lambda_i^\star)\right]\right.$$

$$+ (1 - \tilde{p}(\tilde{s}, \tilde{q}))\left[1 - \mathcal{R}_\lambda^{i+1}\left(\tilde{s}, \tilde{q} + \frac{1}{n}\right) \exp(-\lambda_i^\star)\right]$$

$$+ \lambda^2 \tilde{p}(\tilde{s}, \tilde{q})(1 - \tilde{p}(\tilde{s}, \tilde{q}))\left[1 - \mathcal{R}_\lambda^{i+1}\left(\tilde{s}, \tilde{q} + \frac{1}{n}\right) \exp(\lambda_i^\star)\right]$$

$$\times \left[1 - \mathcal{R}_\lambda^{i+1}\left(\tilde{s} + \frac{1}{n}, \tilde{q} + \frac{1}{n}\right) \exp(\lambda_i^\star)\right];$$

$$- \exp\left(\frac{-\lambda}{2n}\right)\left[\mathcal{R}_\lambda^i(\tilde{s}, \tilde{q}) \exp(\lambda_i^\star)\right]$$

$$\left. \right\}$$
$$\tag{41}$$

Then we have

$$\mathcal{R}_\lambda^i(\tilde{s}, \tilde{q}) = \exp(\lambda_i^\star) \min\left\{-\tilde{p}(\tilde{s}, \tilde{q}) \exp(-\lambda \tilde{p}(\tilde{s}, \tilde{q})) \mathcal{R}_\lambda^{i+1}\left(\tilde{s} + \frac{1}{n}, \tilde{q} + \frac{1}{n}\right)\right.$$

$$- (1 - \tilde{p}(\tilde{s}, \tilde{q})) \mathcal{R}_\lambda^{i+1}\left(\tilde{s}, \tilde{q} + \frac{1}{n}\right) + \lambda^2 \tilde{p}(\tilde{s}, \tilde{q})(1 - \tilde{p}(\tilde{s}, \tilde{q})) \mathcal{R}_\lambda^{i+1}\left(\tilde{s}, \tilde{q} + \frac{1}{n}\right)$$

$$\times \mathcal{R}_\lambda^{i+1}\left(\tilde{s} + \frac{1}{n}, \tilde{q} + \frac{1}{n}\right); - \exp\left(\frac{-\lambda}{2n}\right)\left[\mathcal{R}_\lambda^{i+1}(\tilde{s}, \tilde{q})\right]$$

$$\left. \right\}$$
$$\tag{42}$$

Finally, by selecting the parameters appearing in equation 15, $\delta_t = \delta_q = \delta_s = \frac{1}{n}$, based on the proof of Theorem 1, we derive a numerical schema that satisfies the following PDE equation 27, which corresponds to the two-armed Bernoulli Bandit setting and we achieve the proof by giving the limit of the optimal policy.

# B  Numerical experiments

## B.1  Example 5.1

We give some numerical experiments for the Bernoulli case, our equation is considered to be more general and it takes into account the $\lambda$ variation, more specific when $\lambda$ goes to 0 our equation is exactly the one obtained in Zhu et al. (2022) for a reward value function. In other words, the above equation proves that the rescaled value function $\mathcal{R}_\lambda^i$, when $n$ goes to $\infty$, satisfies the above HJB equation with quadratic growth.

In the next step, we outline the methodology for implementing a numerical solution to the given equation in a computational setting, highlighting the steps necessary to achieve the desired results. The equation equation 27 involves partial derivatives $(\partial_t, \partial_s, \partial_q, \Delta_s)$ which necessitates the use of an appropriate numerical technique, such as the finite difference method, to obtain the solution. Therefore, using the finite difference method, obtain the figure that represents the final solution of the equation equation 27 for a different value of $\lambda$ as follows Figure 1, illustrates the evolution of the value function $\mathcal{R}_\lambda$ as a function of the parameters $(s, q)$ and for a different value of $\lambda$. The plots are obtained for $\lambda = 0$ and for this case we can see that the solution $\mathcal{R}_0(s, q)$ is not defined over the whole domain. In contrast, for $\lambda > 0$ and takes values such 1 and 5 we can see that the regret reaches a minimum value 0 except at $s = q = 1$ it takes negative values. This is clearly depicted in Figure 3, where the 3D representation of the solution is shown. Both Figures are obtained for the following data: $Nt = 100, Ns = 100 Nq = 100$, We can also observe that the value function $\mathcal{R}_\lambda$ takes on negative values, expressed as powers of 10. This occurs because, for a maximum cumulative reward $\omega^i$ and for $\lambda > 0$, the regret expression given by equation 35 converges to 0.

Later we represent the practical performance of the explicit approximate optimal policy for the Two bandit problems described above and compare it with the well known policies such as UCB and Thompson policy.

## B.2 FIGURES

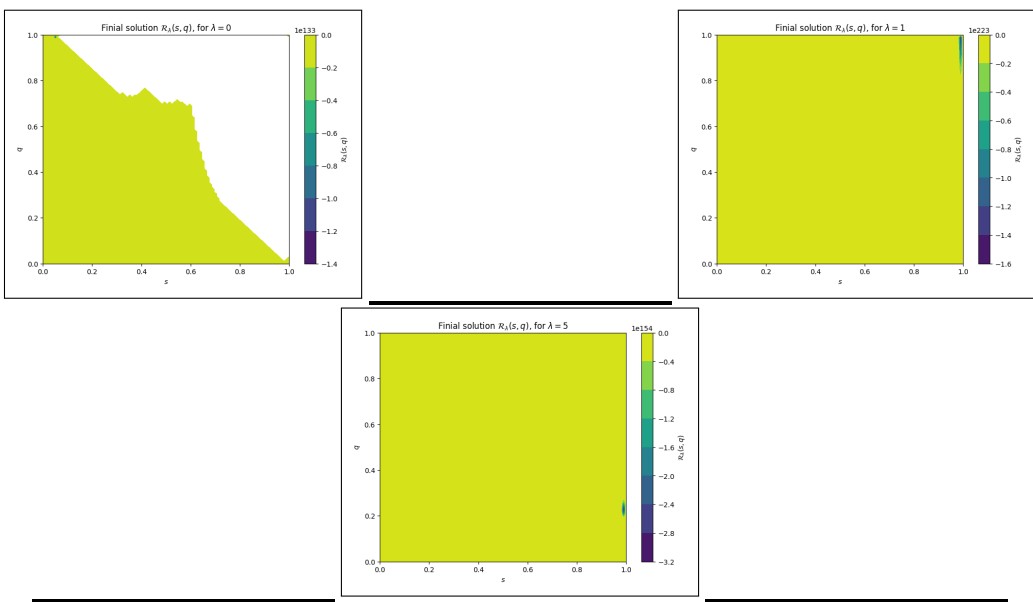

Figure 1: The rescaled value function $\mathcal{R}_\lambda$, for a different value of $\lambda$

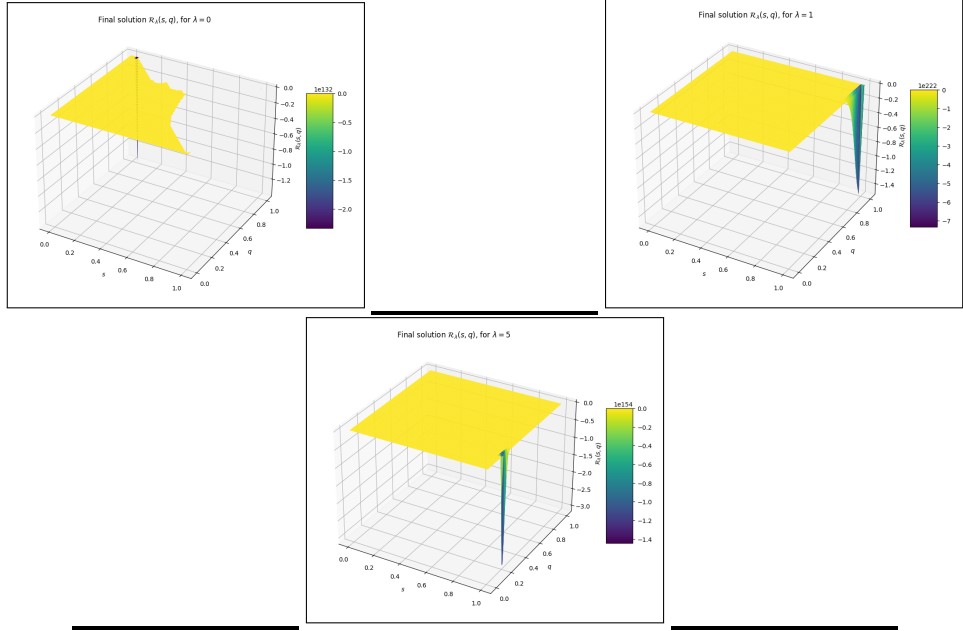

Figure 2: The rescaled value function in 3D $\mathcal{R}_\lambda$, for a different value of $\lambda$

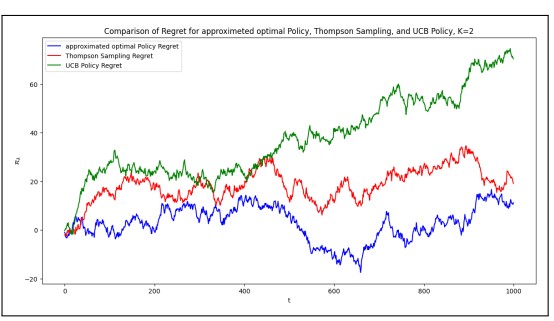

Figure 3: The plot shows the regret $\mathcal{R}_\lambda$ in the 2-armed bandit problem for the approximate optimal policy, UCB and Thompson sampling.

