# OpenReview forum: "Regret measure in continuous time limit for a stochastic Multi-armed bandit problem"
_ICLR.cc/2025/Conference — Submitted to ICLR 2025_

### Official Review · Reviewer_gWB4 · 2024-10-25

**Soundness:** 2
**Presentation:** 1
**Contribution:** 2
**Rating:** 3
**Confidence:** 4

**Summary:**

This paper considers the traditional multi-armed bandit problem with a new risk measure. The authors continuize the time through rescaling and use PDE to find the optimal policy. In the meantime, the authors use some simulations to verify their results.

**Strengths:**

The way to convert the MAB problem to a PDE problem is interesting and meaningful. The work compares different concepts, like frequentist and Bayesian settings making it easy to understand the applicability of the method.

**Weaknesses:**

1. The writing needs to be improved. There are a lot of typos which make it hard to understand the paper.

2. There are no real-world applications provided by the author regarding why this new risk measure is important, reducing the credibility and impact of the paper.

3. The usage of MDP seems improper. In your setting, $\nu$ seems to be fixed and only $s$ and $q$ are changing. However, there is no need to learn the transition kernel as if you choose an action $a$, corresponding $q$ will be increased by 1. Then, it reduces to learning the reward function which is the same as in traditional MAB literature and so people usually don't call it MDP. It's more reasonable to use your framework to consider the case that $\nu$ is varying and say it's MDP.

4. The notations are messy. For example, why $V_{i+1}$ only relies on $R_i$? And you use a very strong assumption but only hide it in the Lemma 1.

5. The Theorem 1 is unclear. What is zero? Why do you use a bracket but link it to nothing?

6. In your numerical study, how do you implement UCB and TS? Do you adjust their definitions of regrets to your new risk measure? If not, they are not comparable. Otherwise, it's better to mention how you set the baseline in detail.

**Questions:**

See Weaknesses.

---

### Official Review · Reviewer_Z8Gu · 2024-10-26

**Soundness:** 2
**Presentation:** 1
**Contribution:** 1
**Rating:** 3
**Confidence:** 3

**Summary:**

This paper studies a class of stochastic multi-armed bandit problems with a risk-sensitive regret measure within a continuous limit setting

**Strengths:**

Considering continuous-time limit of regret measures in continuous time.

**Weaknesses:**

The presentation is not clear.

The paper's contribution and the significance of the problem are not clearly articulated in the Introduction and the main text.

The English in the paper could benefit from some further refinement or editing to enhance clarity and coherence.

**Questions:**

1. what is the main contribution of the paper?

2. what is exactly the problem studied?

3. why studying the continuous-time limit is relevant for bandit problems?

4. How should we interpret the main result Theorem 1 and understand its practical relevance?

---

### Official Review · Reviewer_TCho · 2024-10-29

**Soundness:** 1
**Presentation:** 1
**Contribution:** 1
**Rating:** 1
**Confidence:** 1

**Summary:**

This paper aims to analyze multi-armed bandit problems using differential equations and introduces a new risk measure for the analysis.

**Strengths:**

I am unable to provide a comprehensive scientific review of the paper, and thus I cannot identify specific strengths. Please refer to the weaknesses  below.

**Weaknesses:**

The paper has significant issues with presentation. Not only are there numerous grammatical errors, typos, and punctuation mistakes, but many sentences are incomplete and seem disconnected from the surrounding context. Additionally, the writing lacks a clear logical flow, making it difficult to follow the argument.

Furthermore, it appears that the authors have not adhered to the official ICLR style guidelines.

Due to these issues, I am unable to provide a more detailed review.

**Questions:**

Please refer to the weaknesses.

---

### Meta-Review · Area_Chair_DrG3 · 2024-12-16

**Metareview:**

The reviewers struggled with identifying the high-level approach of this paper. The ratings are overly harsh in my opinion but the paper definitely needs a thorough revision to get it into a publishable state. That includes simple things like polishing the text, but it also would help to motivate and highlight the contributions of the paper early on.

**Additional Comments On Reviewer Discussion:**

Unfortunately, no rebuttal was submitted by the authors.

---

### Decision · Program_Chairs · 2025-01-22

Reject